# Following Ancestral Footsteps: Co-Designing Agent Morphology and Behaviour with Self-Imitation Learning

Sergio Hernández-Gutiérrez
Aalto University
Finland
sergio.hernandezgutierrez@aalto.fi

Ville Kyrki
Aalto University
Finland
ville.kyrki@aalto.fi

Kevin Sebastian Luck
Vrije Universiteit Amsterdam
The Netherlands
k.s.luck@vu.nl

*Abstract*—In this paper we consider the problem of co-adapting the body and behaviour of agents, a long-standing research problem in the community of evolutionary robotics. Previous work has largely focused on the development of methods exploiting massive parallelization of agent evaluations with large population sizes, a paradigm which is not applicable to the real world. More recent data-efficient approaches utilizing reinforcement learning can suffer from distributional shifts to transition dynamics as well as to states and action spaces when experiencing new body morphologies. In this work, we propose a new co-adaptation method combining reinforcement learning and State-Aligned Self-Imitation Learning. We show that the integration of a self-imitation signal improves data-efficiency, behavioural recovery for unseen designs and performance convergence.

## I. Introduction

Finding an optimal combination of body and morphology of agents has been a long-standing research problem, finding its roots in the community of evolutionary robotics [17, 7, 8]. Originally, research in this area largely focused on the use and development of evolutionary or genetic algorithms adapting body and control parameters at the same time [17, 27, 2, 5, 15]. More recent research [13, 19] has studied the benefits of considering the different time-scales on which co-adaptation of body and behaviour occurs in the real world: adaptation of the body is costly and time-consuming, as it involves growing appendices, organs and tissue in nature; likewise in robotics, where even fast manufacturing methods like 3D-printing require a considerable amount of work-hours and material.

Recent years have brought forward several works considering the use of reinforcement learning (RL) methods for the problem of co-adapting robots [6, 21, 25, 19], usually with fast behavioural learning and slower morphology adaptation. This allowed to develop methods capable of being deployed on real-world robotics due to their data-efficiency. However, data-efficient co-adaptation processes can suffer considerably from the problem of distributional shift inherent to the co-adaptation problem setting. Every new agent morphology the algorithms experiences brings with it changes to the transition distribution, as well as to the semantics of state and action spaces.

We propose a novel co-adaptation methodology tackling the aforementioned problems by combining reward-driven RL and self-imitation learning (SIL) utilizing Wasserstein distances for data-efficient adaptation of body and behaviour. Our approach not only forces the RL algorithm to adapt body and behaviour for maximizing an objective function, but also to encourages the imitation of the agent's own previous behaviours to increase learning stability and accelerate the learning progress.

In this paper, we present the following contributions:
**(C1)** An extension of State-Alignment Imitation Learning (SAIL) [18] for mismatching morphologies to State-Aligned Self-Imitation Learning for the problem of co-adapting the morphology and behaviour of agents.
**(C2)** A novel co-adaptation method, **Co**-Adaptation with **S**elf-**I**mitation **L**earning (CoSIL), utilizing State-Aligned Self-Imitation Learning to optimize an agent's morphology and behaviour data-efficiently on fewer design iterations.
**(C3)** We empirically demonstrate the benefits and limitations of a SIL signal by evaluating CoSIL versus a non-self-imitating baseline in a range of locomotion tasks.

## II. Background

*a) Multi-Body Reinforcement Learning:* We consider an extension to the classic Markov Decision Process (MDP) suitable for modelling the fact that both behaviour and morphological parameters are adapted. The Multi-Body MDP (MB-MDP) consists of $(S, A, \Xi, r, p(s_{t+1}|s_t, a_t, \xi), p(s_0))$ with state space $S \in \mathbb{R}^s$ and action space $A \in \mathbb{R}wa$. Notably, in a MB-MDP the set $\Xi$ models the morphological parameter space, containing individual instances of agent morphologies $\xi \in \Xi$. Throughout this paper, we will without a loss of generality consider $\Xi \in \mathbb{R}^d$ for $d$ continuous design parameters. As changes to the agent morphology impact its dynamics, the transition function $p(s_{t+1}|s_t, a_t, \xi)$ depends on the current morphology parameter $\xi$. The reward function $r(s_t, a_t, \xi)$ may also implicitly depend on $\xi$ via the transition function, or explicitly if the manufacturing costs are taken into account, for example. The objective is to find a policy $\pi_\theta(s_t, \xi) = a_t$ which maximizes the finite-horizon expected discounted reward

$$R(\xi, \pi) = \mathbb{E}_{\substack{s_{t+1} \sim p(s_{t+1}|s_t, a_t, \xi) \\ s_0 \sim p(s_0|\xi) \\ a_t \sim \pi(s_t, \xi)}} \left[ \sum_{t=0}^{T} \gamma^t r(s_t, a_t, \xi) \right] \quad (1)$$

given an embodiment $\xi$, the policy $\pi$, and discount factor $\gamma \in (0,1)$.

*b) Co-Adaptation of Agent Body and Behaviour:* The previous formalism allows us to formulate the joint optimization of behaviour and morphology of agents as

$$\pi^*, \xi^* = \arg \max_{\xi} \max_{\pi} R(\xi, \pi); \qquad (2)$$

in other words, we are interested in finding both the optimal morphology $\xi^*$ and optimal policy $\pi^*$ given a reward function $r(s_t, a_t, \xi)$. If we consider the semantics of the parameters and the optimization time-scales (i.e., policy learning can be done faster than morphology adaptation), this problem can be considered a bi-level optimization problem. Given the current morphology of the agent in the inner optimization problem, we can solve the RL problem using Eq. (1). In the outer optimization problem, given performances $R(\xi, \pi)$ of past morphology-policy pairs $(\xi_i, \pi_i)$, we can again utilize optimization methods or reinforcement learning to find new candidate morphologies $\xi$ to evaluate.

## III. CO-ADAPTATION WITH SELF-IMITATION LEARNING

In this section, we will first introduce the individual components of ***Co-Adaptation with Self-Imitation Learning (CoSIL)*** using State-Aligned Imitation Learning (SAIL) [18]. We will end the section with a description of the main algorithm.

### A. Self-Imitation Learning on Co-Adaptation Sequences

Assume a MB-MDP $(S, A, \Xi, r, p(s_{t+1}|s_t, a_t, \xi), p(s_0))$, as given in Section II-0a. Naturally, a co-adaptation process will produce a sequence of morphology-policy tuples $\{(\xi_0, \pi_0), (\xi_1, \pi_2), (\xi_3, \pi_3), \cdots\}$. Given two morphology-policy pairs $(\xi_i, \pi_i)$ and $(\xi_j, \pi_j)$, we can formulate the trajectory distributions

$$q(\tau^i) = p(s_0|\xi_i) \prod_{t=0}^{T-1} p(s_{t+1}|s_t, a_t, \xi_i) \pi_i(a_t|s_t, \xi_i) \qquad (3)$$

and

$$p(\tau^j|\pi_j) = p(s_0|\xi_j) \prod_{t=0}^{T-1} p(s_{t+1}|s_t, a_t, \xi_j) \pi_j(a_t|s_t, \xi_j). \qquad (4)$$

We will now assume that the pair $(\xi_i, \pi_i)$ represents our expert. If we are now currently training on morphology $\xi_j$, where $j > i$, then we can force the policy $\pi_j$ to imitate the previous agent by optimizing

$$\min_{\pi_j} \mathcal{D}(q(\tau^i), p(\tau^j|\pi_j)), \qquad (5)$$

for a divergence measure $\mathcal{D}$ expressing the distance between these two probability distributions. Importantly, we consider here that $\xi_j$ is fixed and not optimized, otherwise $(\xi_i, \pi_i)$ is a trivial solution to this problem. While different choices exist for this divergence measure, we will follow state alignment-based imitation learning and use state-distribution matching via generative adversarial learning.

### B. Feature-State-Distribution Self-Imitation Learning

As previously described, a core problem for imitation learning (IL) between agents with different morphologies is that the semantic of state and action spaces can shift considerably. Hence, using the original state and action spaces is not suitable to use in the IL setting. Therefore, we assume in the following a function $\phi : S \to S^F$ which maps the state of the agent to a shared feature space $S^F$ modelled with motion capture markers placed on the robot's body.

In our proposed SIL approach for co-adaptation, we are matching the state distributions between previous expert behaviour and the current agent, a technique used successfully in prior work [9, 22]. Similarly, we use the marginal feature-space state distributions for the expert trajectories from past morphologies

$$q(\phi(s)) = \mathbb{E}_{\substack{s_{t+1} \sim p(s_{t+1}|s_t, a_t, \xi_i) \\ a_t \sim \pi_i(a_t|s_t, \xi_i) \\ s_0 \sim p(s_0|\xi_i)}} \left[ \frac{1}{T} \sum_{t=0}^{T} \mathbb{1}(\phi(s_t) = \phi(s)) \right] \qquad (6)$$

and for the current agent morphology

$$p(\phi(s)|\pi_j) =$$
$$\mathbb{E}_{\substack{s_{t+1} \sim p(s_{t+1}|s_t, a_t, \xi_i) \\ a_t \sim \pi_i(a_t|s_t, \xi_i) \\ s_0 \sim p(s_0|\xi_i)}} \left[ \frac{1}{T} \sum_{t=0}^{T} \mathbb{1}(\phi(s_t) = \phi(s)) \right], \qquad (7)$$

with $\mathbb{1}$ being a Kronecker delta function, returning the value 1 iff $\phi(s_t) = \phi(s)$ holds true and 0 otherwise. Using these state distributions we can now reformulate Eq. (5) with

$$\mathcal{D}(q(\phi(s)), p(\phi(s)|\pi_j)), \qquad (8)$$

where we can use divergences such as Kullback-Leibler's, the Wasserstein distance, or the Jensen-Shannon divergence. Eq. (8) will be our main objective for enabling SIL across morphologies.

### C. Imitation Reward and Environmental Reward

CoSIL makes use of two reward functions: $r^{\text{RL}}$ for the environment reward we aim to maximize as the main objective and $r^{\text{IL}}$ for the SIL reward, given a demonstration dataset $\tau^{\text{E}}$. We implement $r^{\text{IL}}$ was State-Aligned Imitation Learning (SAIL) using the Wasserstein distance [18] with reward function

$$r^{\text{IL}}(\phi(s_t), \phi(s_{t+1})) = \rho(\phi(s_{t+1})) - \mathbb{E}_{s \sim \tau^{\text{E}}}[\rho(\phi(s))], \quad (9)$$

where $\rho$ is a learned discriminator function (i.e., a neural network) modelling the Kantorovich's potential, assigning higher values to states similar to those seen in the expert dataset $\tau^{\text{E}}$.

### D. Policy Learning with Self-Imitation Learning

CoSIL makes use of Soft Actor Critic (SAC) [12] as the reinforcement learning backbone of the method with a slight adaptation to the learning rule for policy updates. As we have

two reward functions, we propose to adapt SAC to learn two Q-functions with

$$\mathcal{L}_{Q_k^{\mathrm{RL}}} = \frac{1}{2}(Q_k^{\mathrm{RL}}(s_t, a_t, \xi) - (r^{\mathrm{RL}}(\phi(s_t), \phi(s_{t+1})) +$$
$$\gamma(\min_{k=1,2} Q_k^{\mathrm{RL}}(s_{t+1}, a_{t+1}, \xi) - \alpha \log(\pi(a_{t+1}|s_{t+1}, \xi)))))^2, \tag{10}$$

$$\mathcal{L}_{Q^{\mathrm{IL}}_k} = \frac{1}{2}(Q_k^{\mathrm{IL}}(s_t, a_t, \xi) - (r^{\mathrm{IL}}(\phi(s_t), \phi(s_{t+1})) +$$
$$\gamma(\min_{k=1,2} Q_k^{\mathrm{IL}}(s_{t+1}, a_{t+1}, \xi) - \alpha \log(\pi(a_{t+1}|s_{t+1}, \xi)))))^2. \tag{11}$$

To avoid imbalances during training, we normalize both rewards using z-score normalization. This leads to the following loss function for the policy $\pi$ with two Q-networks:

$$\mathcal{L}_\pi = (1 - \omega) \min_{k=1,2} Q_k^{\mathrm{RL}}(s_t, a_t, \xi) +$$
$$\omega \min_{k=1,2} Q_k^{\mathrm{IL}}(s_t, a_t, \xi) - \alpha \log \pi(a_t \mid s_t, \xi), \tag{12}$$

in which we optimize the policy for both the objective-driven Q-function $Q_{\mathrm{RL}}$ and the SIL Q-function $Q_{\mathrm{IL}}$, weighted by $\omega$. Both critics use the double-Q trick proposed by [14].

### E. Morphology Optimization

Similar to the behaviour learning process, we extend the morphology optimization objective to incorporate SIL. Accordingly, we supplement the objective introduced in [19] by adding the Q-function $Q_j^{\mathrm{IL}}$ with

$$\max_{\xi} \mathop{\mathbb{E}}_{s_0 \sim p(s_0|\xi)} [(1 - \omega_{\mathrm{opt}}) \min_{j=1,2} Q_j^{\mathrm{RL}}(s_0, \pi_{pop}(a_0|s_0, \xi), \xi) +$$
$$\omega_{\mathrm{opt}} \min_{j=1,2} Q_j^{\mathrm{IL}}(s_0, \pi_{pop}(a_0|s_0, \xi), \xi)], \tag{13}$$

where $\omega_{\mathrm{opt}}$ is used to weigh the importance of the SIL reward versus the environment reward function. While in principle any optimization method can be used, we found the gradient-free Particle Swarm Optimization (PSO) optimizer [16] to be the most efficient. Since the distribution $p(s_0|\xi)$ is generally unknown, we replace it in practice with $s_0 \sim R_0$, where $R_0$ is a replay buffer containing only starting states.

## IV. EXPERIMENTS

In order to understand the impact of a SIL signal in the co-adaptation setting, we seek to empirically answer the following research questions:

**(RQ1)** Does SIL help to better co-adapt the behaviour and morphology of agents?
**(RQ2)** Can SIL help to counter distributional shifts and recover faster after a morphology change?
**(RQ3)** What are the limitations of SIL? Is it always beneficial?

### A. Experimental Setup

In our experiments, we used variants of the OpenAI Gym library [4] environments Humanoid and HalfCheetah adapted to the co-adaptation setting, as previously proposed [22], and also evaluate on a new adaptable variant of the Walker environment. These environments are implemented using the MuJoCo engine

[26]. Experiments are conducted on a computing computer with Nvidia RTX A4500. We employed 32GB of RAM and were constrained by 72 hours of real time usage per experiment. The results are averaged across four distinct seeds. In all experiments, we compare CoSIL to a baseline which does not include a SIL signal neither in the critic's update function nor in the design evolution objective function. In this way, we can study the effects of a SIL signal in isolation from other performance factors (RQ1).

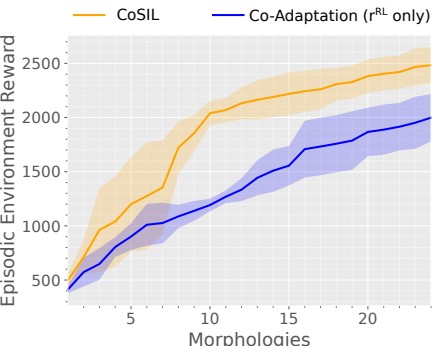

(a) Humanoid (300 episodes per design)

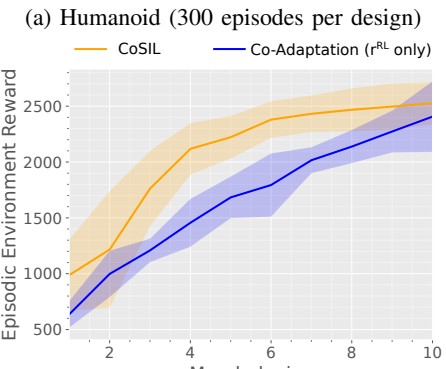

(b) Humanoid (1000 episodes per design)

Fig. 1: The cumulative episodic rewards of CoSIL (orange) and the baseline (Co-Adaptation, blue) in the Humanoid task. Both figures show the average performance of morphologies in terms of environmental rewards, by averaging the episodic return of the best 20% of episodes. Experiments were repeated four times, and morphologies are sorted by their performance along the x-axis. Figure (a) shows the Humanoid task in which each morphology was trained for 300 episodes, while morphologies in Figure (b) were optimzed every 1000 episodes. It can be seen that the proposed method (CoSIL) shows better performance than using solely the environmental reward (Co-Adapt), with comparable perfomance when trained for 300 and 1000 episodes.

The goal of the agent in the morphology-adaptable Humanoid task is to learn a policy allowing stable but fast forward locomotion, as proposed in [22]. To achieve this, the following objective function $r^{\mathrm{RL}}$ is used:

$$r_t^{\mathrm{Humanoid}} = 1.25(c_t - c_{t-1}) - 0.1\mathrm{ctrl}_t^2$$
$$- \min(0.5 \times 10^{-6}\mathrm{cfrc\_ext}_t^2, 10) + 5, \tag{14}$$

where $c_t$ is the position of the center of mass of the robot

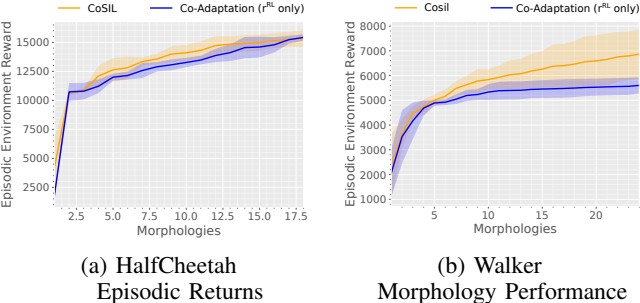

(a) HalfCheetah
Episodic Returns

(b) Walker
Morphology Performance

Fig. 2: The cumulative episodic rewards of CoSIL (orange) and the baseline (Co-Adaptation, blue) in the HalfCheetah and Walker task. Figures show the average return and standard deviation of the environmental reward, averaged over four seeds, and with morphologies sorted by their performance (worse performing morphologies left, better performing morphologies right). While using self-imitation for co-adaptation (CoSIL) shows considerably better performance in Walker (b), we found the performance gain in HalfCheetah (a) to be negligible. We hypothesise this is due to the simplicity of the HalfChetah task.

at timestep $t$, $\text{ctrl}_t$ are the actuator activations at timestep $t$ and $\text{cfrc\_ext}_t$ are the external forces acting on the body of the robot at timestep $t$. In this experiment, we compare CoSIL (red) using both $r^{\text{RL}}$ and $r^{\text{IL}}$ versus the standard Co-Adaptation baseline using only SAC and the objective reward $r^{\text{RL}}$ (blue). We adapt morphology and behaviour over 10,000 episodes, or 10 selected morphologies. For CoSIL, a weight-parameter of $\omega = \omega_{\text{opt}} = 0.2$ was used, which was determined via hyper-parameter optimization through grid-search. From the results in Fig. 1b, it can be seen that CoSIL outperforms standard Co-Adaptation without SIL after a handful of designs. Fig. 1a shows a comparison between CoSIL and Co-Adaptation when training on each morphology for only 300 episodes, which highlights the ability of CoSIL to find better performing morphology-policy combinations in a sample-efficient manner.

### B. Limitations of CoSIL

Next, we investigate possible limitations of CoSIL over reward-driven co-adaptation (RQ3). For this, we evaluate CoSIL on the environments of varying difficulty, specifically the relatively simple HalfCheetah task and the harder Walker task. We selected two variations of the HalfCheetah locomotion task in MuJoco with the reward function

$$r_t^{\text{RL}} = \max(\frac{1.25}{\Delta t} \cdot (c_t - c_{t-1}) - 0.1 \sum_i \mathbf{a}_{t,i}^2, 0), \qquad (15)$$

where $\mathbf{a}_{t,i}$ is the $i$-th action taken at timestep $t$ and $c_t$ is the position of the center-of-mass at time step t, and $\Delta t$ the time difference. For Walker we use the reward function

$$r_t^{\text{RL}} = \max(\frac{1}{\Delta t} \cdot (c_t - c_{t-1}), 0) - 0.1 \cdot |\mathbf{a}|_2 - 0.1 \cdot |\theta|, \quad (16)$$

where $\theta$ is the Euler angle of Walker's orientation in radians. In both tasks, we optimize the leg-segment lengths of the agent, thus resulting in 6 morphology parameters for HalfCheetah and 4 for Walker. We use again $\omega = \omega_{\text{opt}} = 0.2$ for both

HalfCheetah and Walker. As we can see in Fig. 2b/d, CoSIL outperforms standard Co-Adaptation using only $r^{\text{RL}}$ in the Walker task, as also observed in the Humanoid task. However, we note that CoSIL and Co-Adaptation show a very similar performance during the first designs. The performance gap between both approaches closes further if we compare them on the simpler HalfCheetah task. Fig. 2 (a) clearly show that using a SIL reward $r^{\text{IL}}$ is not helpful in this case. Additionally, this could indicate that the distributional shift of state and action spaces in low-complexity tasks is lower, as one would expect. From this, we can conclude that, especially on simpler and less complex co-adaptation tasks, the additional use of SIL should not be preferred, especially given that CoSIL requires in total more compute due to the deep imitation learning methods used.

## V. RELATED WORK

**Evolutionary Robotics:** Designing robot hardware with evolutionary principles has been a long-standing research effort. Seminal work by [17] explored using genetic algorithms to co-adapt a simple controller. Earlier works by [24] used competition as a reward in a genetic algorithm to adapt the bodies of two robots in a fighting task. Approaches for evolutionary robotics have been successfully applied in simulation [3], although recent works have identified that applying them in real world scenarios remains an open challenge [8]. Recent work has focused primarily on the fast changeability of robotic platforms as means to allow real world evolution of robots [20, 13, 1], although this constrains the range of possible robot designs considerably.

**Co-Adaptation with Reinforcement Learning:** Recent works on co-adaptation have sought to improve data-efficiency and feasibility by using a RL method as its main component. Seminal work by [11] employs REINFORCE to jointly co-adapt the body and behaviour of agents [28]. [23] extended this approach by proposing a deep RL co-adaptation algorithm. Increased data-efficiency was achieved by [19] with the introduction of an off-policy deep RL method using the Q-value function for design evaluations. Another recent work [10] employed deep RL with mass-parallelization of agent populations in simulation.

## VI. CONCLUSION

We presented a new co-adaptation method named **Co**-Adaptation with **S**elf-**I**mitation **L**earning (CoSIL) which introduces the idea of using a SIL reward within a reward-driven co-adaptation framework using deep reinforcement learning. To achieve this, we used State-Aligned Imitation Learning (SAIL) [18], introduced a method to select and match expert data from previously seen morphology-policy combinations, and employed separate Q-value functions for the objective and imitation rewards to increase data-efficiency when optimizing the morphology parameters. In experiments on morphology-adaptable agents in simulation, we showed that by imitating previously seen behaviour we can combat the distributional shift in dynamics, action and state spaces, as well as recover

faster when switching to a newly selected agent morphology. However, CoSIL requires a larger amount of computational effort due to additional deep neural network training, which makes it not preferable for simple co-adaptation problems. Nevertheless, with the methodology proposed in this paper we make a further step towards the useful integration of imitation learning techniques into co-adaptation techniques. Several interesting avenues for future work are opened up by our work, such as the use of quality-diversity approaches for selection of self-demonstrations, or further investigations of using a SIL reward during design optimization.

## ACKNOWLEDGEMENTS

This work was supported by the Research Council of Finland Flagship programme: Finnish Center for Artificial Intelligence FCAI. Kevin Sebastian Luck is supported by project number NGF.1609.241.015 of the research programme AiNed XS Europe which is financed by the Dutch Research Council (NWO) . We acknowledge the computational resources provided by the Aalto Science-IT project.

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
