# OpenReview forum: "Following Ancestral Footsteps: Co-Designing Agent Morphology and Behaviour with Self-Imitation Learning"
_roboticsfoundation.org/RSS/2024/Workshop/EARL — EARL 2024 Poster_

### Official Review · Reviewer_TW4z · 2024-06-22
**Interesting method for co-adapting the morphology and policy of agents**

**Rating:** 8
**Confidence:** 4

**Review:**

Summary: This paper proposes and evaluates a new method, Co-SIL, for co-adapting the morphology and policy of agents by combining reinforcement learning and state-aligned self-imitation learning.

Strengths:
-	The paper is aligned with the workshop topic very well
-	The proposed Co-SIL method is novel
-	The experiments are conducted with clear research questions
-	Co-SIL achieves higher performance than the baseline which utilizes only environmental reward
Weaknesses:
-	The contributions C1 and C2 appear to be the same. The authors need to distinguish them better
-	The authors mention that Co-SIL is computationally more expensive, some numerical estimates would be helpful
-	In Fig. 1 and 2, the morphologies on x-axis are sorted by their performance, but this misses the information about the sequence in which they were selected
-	The authors should explicitly state which experiment answers the RQ2, just like they state it for RQ1 and RQ3.
-	In the Multi-body MDP, why aren’t the state and action spaces functions of the morphology?

---

### Decision · Program_Chairs · 2024-06-24

Accept (Poster)